# Evaluation of the discriminatory potential of antibodies created from synthetic peptides derived from wheat, barley, rye and oat gluten

**David Poirier** [1,2], **Jérémie Théolier**[1,2], **Riccardo Marega**[3], **Philippe Delahaut**[3], **Nathalie Gillard**[3], **Samuel Benrejeb Godefroy**[1,2]*

**1** Department of Food Science and Nutrition, Pavillon Paul-Comtois, Université Laval, Québec, Québec, Canada, **2** Institute of Nutrition and Functional Foods, Université Laval, Québec, Québec, Canada, **3** Analytical Laboratory, CER Groupe, Marloie, Belgium

* samuel.godefroy@fsaa.ulaval.ca

**Data Availability Statement:** All relevant data are within the manuscript and its Supporting information files.

## Abstract

Celiac disease (CD) is triggered by ingestion of gluten-containing cereals such as wheat, barley, rye and in some cases oat. The only way for affected individuals to avoid symptoms of this condition is to adopt a gluten-free diet. Thus, gluten-free foodstuffs need to be monitored in order to ensure their innocuity. For this purpose, commercial immunoassays based on recognition of defined linear gluten sequences are currently used. These immunoassays are designed to detect or quantify total gluten regardless of the cereal, and often result in over or underestimation of the exact gluten content. In addition, Canadian regulations require a declaration of the source of gluten on the label of prepackaged foods, which cannot be done due to the limitations of existing methods. In this study, the development of new antibodies targeting discrimination of gluten sources was conducted using synthetic peptides as immunization strategy. Fourteen synthetic peptides selected from unique linear amino acid sequences of gluten were bioconjugated to *Concholepas concholepas hemocyanin* (CCH) as protein carrier, to elicit antibodies in rabbit. The resulting polyclonal antibodies (pAbs) successfully discriminated wheat, barley and oat prolamins during indirect ELISA assessments. pAbs raised against rye synthetic peptides cross-reacted evenly with wheat and rye prolamins but could still be useful to successfully discriminate gluten sources in combination with the other pAbs. Discrimination of gluten sources can be further refined and enhanced by raising monoclonal antibodies using a similar immunization strategy. A methodology capable of discriminating gluten sources, such as the one proposed in this study, could facilitate compliance with Canadian regulations on this matter. This type of discrimination could also complement current immunoassays by settling the issue of over and underestimation of gluten content, thus improving the safety of food intended to CD and wheat-allergic patients.

**Funding:** DP Grant number: IT12512 MITACS and r-biopharm Canada inc. https://www.mitacs.ca/ https://r-biopharm.com/ NO The funders had no role in study design, data collection and analysis, decision to publish, or preparation of the manuscript.

**Competing interests:** Samuel Benrejeb Godefroy's research activities are funded by the Ministry of Agriculture, Fisheries, and Food, Government of Quebec, Canada; the Ministry of Science, Technology, and Innovation, Government of Quebec; Canada's Innovation Foundation; the U.S. Department of Agriculture Foreign Agriculture Service; r-Biopharm GmbH; and r-Biopharm Canada Inc. Samuel Benrejeb Godefroy acts as an expert advisor for members of the food and beverage industry, international organizations (the Food and Agriculture Organization of the United Nations, the United Nations Industrial Development Organization, and the World Bank), and international food regulators such as the China National Centre for Food Safety Risk Assessment and consumer organizations such as Food Allergy Canada. The other authors declare that they have no conflicts of interest.

## Introduction

Celiac disease is estimated to affect approximately 1% of the world's population [1]. Symptoms of this condition are triggered by ingestion of cereals containing gluten, such as wheat, barley, rye, and in rare cases oats [2]. To date, adhering to a gluten-free diet is the preferred strategy for the prevention of celiac disease symptoms. A reliable method for gluten quantification is therefore necessary to determine the level of this component in foods. In 1979 the Codex Alimentarius adopted the Standard for Foods for Special Dietary Use for Persons Intolerant to Gluten. Revised in 2008, it states that a food not exceeding a gluten content of 20 mg/kg can be declared as gluten-free [3]. This same standard specifies that the preferred method for the determination of gluten is the R5 Méndez enzyme-linked immunosorbent assay (ELISA). This method has been endorsed by the AOAC as an official method and as a type I method by the Codex Committee of Methods of Analysis and Sampling since 2006 [4, 5]. A type I method "determines a value that can only be arrived at in terms of the method per se and serves by definition as the only method for establishing the accepted value of the item measured" [6].

Since then, several authors have highlighted the harmonization and performance issues of gluten quantification by ELISAs. More specifically for the type I method, it has been shown that the test response varied depending on the type of gluten detected (e.g. barley, rye, wheat) thus leading to over and underestimation [7–10]. Besides, the standard used to calibrate the immunoassay may also impact the results. Most authors agree on the need for better test calibration and better standards [11–19]. However, the use of the most appropriate reference material is still extensively debated [11–19]. The use of a reference material for each grain, namely wheat, barley, rye and oats, as a calibration standard makes it possible to correct or at least reduce this variability [9, 11, 20]. Appropriate calibration with prolamin or glutelin as the analyte has been shown to reduce the discrepancy between measured and actual amounts of gluten [8, 12]. However, this calibration is often impossible in food analysis, since the source of gluten, if present due to cross-contamination, is usually unknown. Therefore, being able to identify the source of gluten in a food sample would help in selecting the appropriate calibration standards. In addition, Canadian legislation requires the declaration of the source of gluten specifying the name of the original grain on the labels of prepackaged foods, which makes the availability of a method to differentiate gluten sources in food samples all the more essential [21]. The ability to distinguish sources of gluten would also increase the food supply of the population suffering from wheat allergy who is deprived of consuming rye, barley and oats because of current analytical limitations. The aim of this study was to create new antibodies capable of distinguishing between different sources of gluten. These new antibodies must therefore target unique linear or conformational epitopes belonging to wheat, barley, rye and oats. The immunization strategy for obtaining the current commercially available antibodies uses native wheat (Skerritt) or rye (R5) prolamins as immunogen. [22, 23]. However, the use of prolamin in its native form confined the reactivity of antibodies to only a few external antigenic sites [24]. G12, in contrast, developed against a 33-mer peptide of α-gliadin, has been identified as a primary initiator of the inflammatory response to gluten in celiac patients [25, 26]. The sensitivity of this monoclonal antibody (mAbs) to wheat, rye, barley and oats does not allow for its use to distinguish different sources of gluten [25]. However, the strategy of immunization using synthetic peptides is promising to obtain antibodies able to differentiate gluten sources. In this study, immunogens were developed by bioconjugating synthetic peptides from respectively unique amino acid sequences of wheat, barley, rye and oats to a carrier protein. Polyclonal antibodies (pAbs) were thus elicited in rabbits, and the evaluation of their relative sensitivity and specificity by indirect ELISA showed their ability to discriminate between different sources of gluten from wheat, barley, rye and oats.

## Materials and methods

### Materials

All chemicals and solvents were HPLC grade as a minimum. Ultrapure water was used for FPLC and buffers (Merck KGaA, Darmstadt, Germany). Grains of wheat cultivar (cv.) AAC harlaka, barley cv. AAC synergy and rye cv. danko were provided by Semican International Inc. (Plessisvile, QC, CA). Oat cv. Ruffian was provided by Avena Foods (Regina, AL, CA). Flours from rice, split pea, chickpea, millet, and soy were purchased from local stores.

### Research and selection of sequences

Gluten protein sequences listed in Table 1 were searched and extracted from the National Center for Biotechnology Information (NCBI) database [27]. For each protein, a multiple alignment of all available sequences was performed using Multiple Sequence Alignment (MSA) tool from *Clustal Omega* [28]. Then, consensus sequences (CS) were obtained for each gluten protein types (GPT) using the sequence editor *Jalview(2.11.0)* [29, 30]. Pairwise sequence alignment was then performed on such CS with every combination of two proteins from Table 1 using *Jalview(2.11.0)* to identify shared and unshared amino acid (a.a.) strands. All unshared sequences from the CS of each grain equal or containing more than 6 a.a. were compiled and kept in a database. Prolamins from maize (zein) [31] and soy (glycinin) [32] were also compared with the CS to discard shared sequences between GPT and these prolamins. The Grand Average of Hydropathy (GRAVY) of the potential candidates as haptens was calculated according to Kyte and Doolittle (1982) [33]. Based on several parameters, such as the consecutive number of the same a.a., the occurrence of the sequence in the GPT and the number of a. a., 14 sequences were selected to produce immunogens.

### Hapten synthesis

To the 14 selected peptides, minor modifications were made to their sequences during their synthesis (Bio Basic Inc., Markam Ontario, Canada). The modifications regard the N-terminal addition of a cysteine (when none was present in the original sequence), in order to introduce a thiol moiety for the subsequent "click-coupling" with protein carriers surfacing maleimido moieties, and N-terminal acetylation / C-terminal amidation to better mimic the polypeptide structure.

### Bioconjugation of immunogen

Each of the 14 synthetic peptides was coupled to hemocyanin from *Concholepas concholepas* (CCH) [34] (Blue Carrier, Biosonda S.A., Santiago, Chile) using the protocol described by

**Table 1. Prolamins and glutelins from gluten-containing grains.**

| Wheat | Barley | Rye | Oat |
|---|---|---|---|
| α/β-gliadins | B-hordeins | γ-secalins | avenins |
| γ-gliadins | B1-hordeins | ω-secalins | |
| ω5-gliadins | B3-hordeins | | |
| ω1,2-gliadins | C-hordeins | | |
| LMW-GS[a] | D-hordeins | | |
| HMW-GS[b] | γ1-hordeins | | |
| | γ3-hordeins | | |

[a]Low-molecular-weight glutenin subunits

[b]High-molecular-weight glutenin subunits

Hermanson et al. (2008) [35] with minor modifications. Briefly, 20 mg of CCH were dissolved in 1 mL of coupling buffer (100 mM sodium phosphate, 0.3M NaCl, pH 7.4) to which were added 2 mg of the heterobifunctional cross-linker, sulfosuccinimidyl 4-[N-maleimidomethyl] cyclohexane-1-carboxylate (sulfo-SMCC) (Thermo Fisher Scientific, MA, USA), previously dissolved in 200 µl of ultrapure water. The mixture was incubated for 30 minutes at room temperature on an orbital shaker (Fisher Scientific, MA, USA). Then, 20 µl of 100 mM glycine in coupling buffer were added to neutralize excess of sulfo-SMCC, followed by purification of the CCH-sulfo-SMCC complexes by FPLC (ÄKTA avant; GE Healthcare, IL, USA) on desalting columns (HiTrap™; GE Healthcare, IL, USA). Ellman's assays were conducted to ensure a good level of maleimide activation of CCH [35, 36], by using L-cysteine as external calibration curve. Maleimide-activated CCH was conjugated by 1.2-fold molar excess of synthetic peptides and allowed to react 1 hour at room temperature, and the peptide loading (nmol peptide/mg CCH) was indirectly obtained upon quantitation of the unreacted thiolated peptide (Ellman assay). The conjugates were snap frozen in liquid nitrogen and stored at -20˚C until further use.

### BSA bioconjugate

Each of the 14 synthetic peptides was coupled to UltraPure™ Bovine Serum Albumin (BSA) (Thermo Fisher Scientific, MA, USA) using the same protocol described above for bioconjugation of CCH, but with 4 mg of Mal-PEG4-NHS (Carbosynth, Compton, UK) as cross-linker.

### Immunization of rabbits

All of the experimentation involving animals was done under the frame of the ethical protocol CE/Sante/E/001 (immunization and production of sera/polyclonal antibodies) approved by the ethical committee of CER Groupe (agreement nb. LA1800104). The agreement LA1800104 was bestowed by the Federal Public Service of the Walloon Region (Belgium). The experimentation respected the legislation in force at the moment of the studies, thus following the guidelines established at the European level (Directive 2010/63/EU revising Directive 86/609/EEC on the protection of animals used for scientific purposes), Belgian level (Arrêté royal relatif à la protection des animaux d'expérience, AR 2013/05/29), and Regional level (Code Wallon du Bien-être animal 03/10/2018). Each CCH-peptide conjugate was injected to three different rabbits for a total of 42 rabbits. Polyclonal antibodies were raised in rabbits by subcutaneous injection of 200 micrograms of CCH-peptide conjugates emulsified with Freund's complete adjuvant for the first injection, or Freund's incomplete adjuvant for all following injections (Becton Dickinson Benelux, Erembodegem, Belgium). Injections were administered on a fortnightly basis and then, from the third injection onward, at the rhythm of one injection every 28 days. Test bleeds were collected 10 days after each immunization (from the third immunization onward). The blood was centrifuged, and the collected serum was stored at -20˚C until used. An aliquot of such serum was diluted 1/10 in a solution of 50% assay buffer (phosphate buffer 65 mM,NaCl 150 mM, 0.2% gelatin, 0.05% Tween 20, 0.01%, 8-anilino-1-naphthalene-sulfonic acid ammonium salt, and ascorbic acid 28 mM) and 50% ethyleneglycol, yielding diluted pAbs solutions that were kept at -20˚C prior to their use for titer and specificity assessment.

### Preparation of gluten protein fractions

Wheat, barley, rye and oat grains were turned into flours with a Grindomix GM 200 (Retsch, Haan, NRW, DE). Flours purity was confirmed by PCR using primers for wheat, barley, rye and oat designed by Sandberg et al. (2003) [37]. Defatting of the flours and extraction of the

different prolamins was performed according to Schalk et al. (2017) [38]. The solvent used for defatting was replaced by hexane [39].

## Titer determination by indirect ELISA

Microtiter plates (F8 Maxisorp™ Nunc-Immuno™ Module, 96-well plates, Thermo Scientific) were coated overnight at room temperature with 1 µg/mL BSA-peptide bioconjugate diluted in 0.05 M carbonate-bicarbonate buffer, pH 9.6 (Fisher Scientific, Pgh, USA). After being washed three times with washing buffer (0.15 M NaCl and 0.05% Tween 20), 250 µl of blocking buffer (PBS with 1% gelatin) were added to each well and the plates were incubated 2 hours at 37˚C. After washing, 100 µl per well of a serial dilution (1:2 000 to 1:256 000 overall) of the previously yielded pAbs solutions in assay buffer were added, along with non-specific binding, and incubated for 1 hour at 37˚C. Plates were washed again and 100 µl per well of anti-Rabbit IgG (whole molecule)–peroxidase antibody produced in goat (Sigma, Saint-Louis, USA, reference# A6154-1ML) diluted 1:10,000 in assay buffer was added. After another three washings step, the chromogenic substrate 3,3′,5,5′-tetramethylbenzidine (TMB, D-Tek, Belgium) was added, and the wells were incubated at room temperature. After 30 minutes in the dark, the reaction was stopped with 1.8 N $H_2SO_4$ and optic density (O.D.) was measured at 450 nm in a microplate reader (Multiskan FC; Thermo Fisher Scientific, MA, USA). Titer value was defined as the highest dilution which triggered an O.D. value of 1.000 at 450 nm.

## Relative sensitivity and specificity tests

The same protocol used for titer determination was used for relative sensitivity and specificity tests against gluten prolamins in native and denatured state, and against rice, split pea, chickpea, millet, and soy flours with modifications regarding the coating of the microtiter plate. Coating with native prolamins was performed by firstly dissolving previously prepared prolamins into 60% (v/v) ethanol at a concentration of 25 mg/mL, and then coating the microplate with a 50 µg/mL native prolamin solution diluted in carbonate-bicarbonate buffer, pH 9.6. Denaturation of prolamins was performed at 25 mg/mL using cocktail solution according to García et al. (2005) [40]. The microplates were coated at 50 µg/mL with a denatured prolamin solution in carbonate-bicarbonate buffer, pH 9.6. The same dilutions of pAbs used for the relative sensitivity tests were used for specificity tests. Results were expressed in intervals of the strongest signal obtained for each serum. The cut-off threshold for discriminating background noise was determined based on the average of several blanks and was statistically fixed at 99.9% confidence level using Frey et al.'s (1998) endpoint titer determination method for immunoassays [41].

## Results and discussion

### Selection of the sequences

The bioinformatic strategy for the selection of sequences to be used as immunogens was based on an MSA combined to a CS (S1 Table). This strategy is the foundation of the results presented in this study and therefore needs to be discussed. Among each GPT, there are often small differences due to substitution, deletion and/or insertion of nucleotides, which contribute to the heterogeneity within each type [38]. Thus, the NCBI database contains a wide range of sequences even within a same GPT. It was therefore important for the present work to be able to separate non-well-preserved regions from well-preserved ones as the former are more representative of the GPT range. When aligning sequences with shared origin such as GPT, the use of MSA allows to effectively identify conserved residues in the dataset [42, 43].

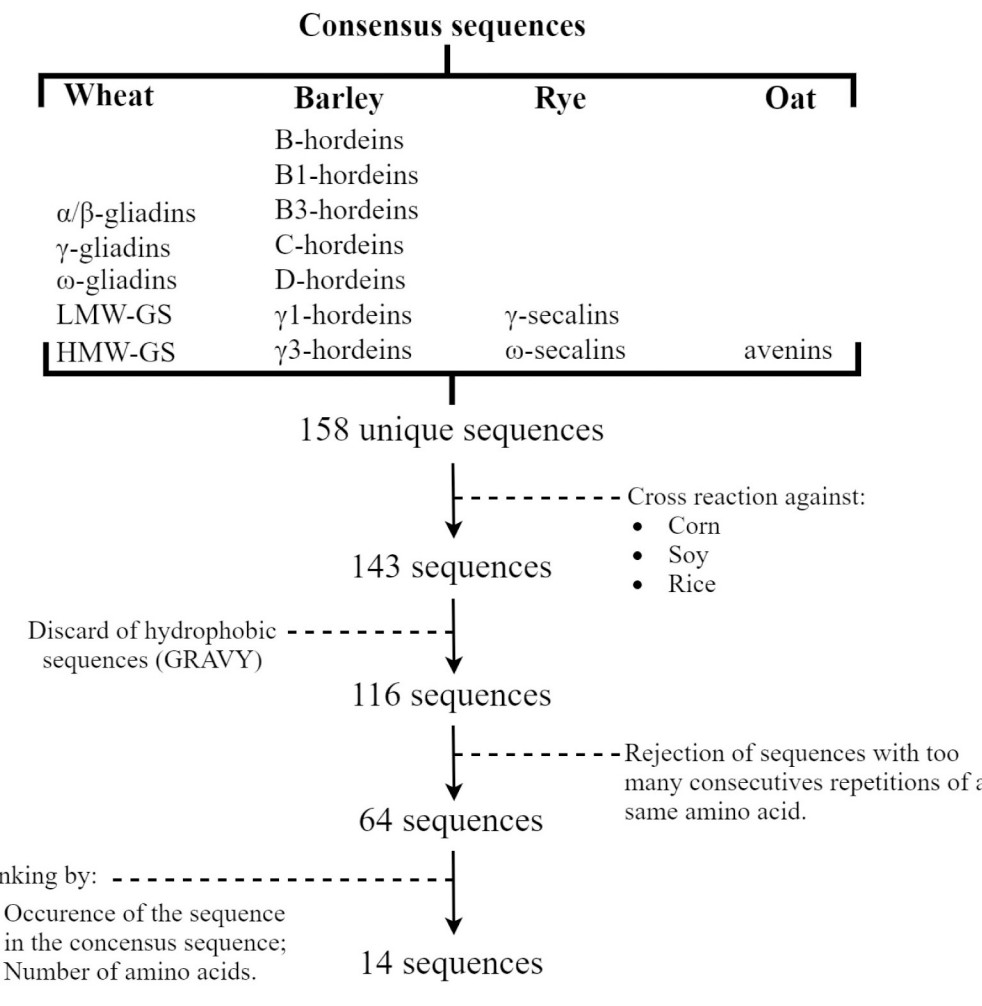

**Fig 1. Peptide selection flow chart.**

Compiling these aligned patterns results in a CS, defined as a compiled sequence of the most common a.a. at each position [29, 44]. The execution of the MSA before the creation of the CS makes the CS model more accurate [29]. It has also been shown that the use of CS to create an immunogen is an effective strategy that minimizes the degree of variable elements within a created immunogen [45]. There are, however, limitations to the use of this bioinformatics strategy. Indeed, a CS is generated independently of the conserved or unpreserved character of the GPT. The quality of the CS directly depends on the quality of the MSA performed upstream. This therefore creates segments of CS showing notable dissimilarity with GPT, and others that are conservative. However, for large data sets, it is difficult to determine which segment of the sequence is from variable or conserved regions, which can result in the selection of less representative synthetic peptides from the GPT. Shared patterns recognized through the pairwise alignment performed on each CS, led to identification of unique sequences for each GPT. Every unshared sequence containing more than 6 a.a., the length that has been showed to consistently elicit antibodies that bind to the original protein, were conserved (158 unique sequences) [46]. The workflow of the subsequent steps is presented in Fig 1. After elimination of cross-reacting sequences with prolamins of corn, soy and rice, the remaining sequences

were divided based on their hydropathy index. Hydrophilic sequences were conserved because of (1) their bioconjugation capacity in biological buffers such as PBS, and (2) their hydrophilic surface-oriented epitopes would be accessible to antibodies [46–50]. However, well-conserved sequences are most often found in the internal structure of the protein molecule and are therefore more hydrophobic [51]. Nevertheless, the authors estimated that the selection of hydrophilic peptides instead of hydrophobic peptides was an adequate strategy for facilitating the creation of the immunogen, and for maximizing the final recognition rate of the antibodies generated, by presenting an antigen to the host with a loose and more linear structure [46–48]. These two constraints make the selection of synthetic peptides challenging. The selection strategy used in this study makes the well-conserved segments of the GPT more reliable. However, the majority of these segments are hydrophobic and therefore less suitable for antibodies production. Conversely, the use of hydrophilic segments as synthetic peptides is less reliable, since they are mostly in less well-preserved segments of the GPT, but is structurally more suitable to raise antibodies. Furthermore, it is known that specificity to an antigen can be highly modified by a change as small as a single a.a. substitution [52]. Thus, the choice of more hydrophilic sequences as immunogen may have caused a decrease in the specificity of the antibodies generated towards the GPT of the respective grains.

The hydropathy index-based selection led to 116 unique sequences. Among them, 52 sequences with too many consecutive repetitions of a same a.a. were rejected for specificity reasons. The remaining 64 sequences were ranked according to their occurrence in their respective GPT's consensus sequences, their number of a.a., and the proportion of their respective GPT in gluten of wheat, barley, rye and oat based on Schalk et al. (2017) data [38]. If the site of the selected peptide is poorly exposed or even criptic, the raised antibody is more likely to be unable to recognize the native protein [53]. Thus, N- or C-termini external, charged and polar regions are often good choices for the selected peptides [49]. Nevertheless, in the present study, the selected peptides firstly aimed to discriminate GPT homologically close, regardless of their conformations, gluten extraction is not always performed under denaturing conditions [40, 54, 55]. Peptides with more than 8 a.a were prioritized to improve recognition of the original protein [48]. On the other hand, peptides longer than 20 a.a were discarded as they may adopt conformations that will no longer resemble the origin protein, leading to poorer specificity [48]. These considerations restrained the choice to 14 sequences (six for wheat, four for barley, three for rye and one for oat) that were selected to produce immunogens to raise pAbs.

## Titer determination

Except for rabbits 24 and 41, which died before the first bleed, titer determinations were performed on the pre immune sera and on test bleed number 4, in order to maximize the occurrence of stable titers levels. It should be pointed out that the peptide-BSA conjugates used for these ELISA assessments further differ from the corrresponding CCH conjugates in light of the different spacer used in the heterobifunctional crosslinker (tetraethyleneglycol instead of cyclohexyl), which ensures that reactive antibodies are specific to the peptides.

Results of the indirect ELISA (Fig 2) against synthetic peptides used for the titer determination reveal titer determinations higher than 1:10 000 for all the sera with the exception of those obtained from immunogens produced with peptides 10 (barley) and 13 (Rye). Very high dilutions (above 1:96 000) were obtained for the immunogens produced with peptides 2 and 5 (wheat), and peptide 9 (barley). Equivalent values could not be obtained for any immunogen containing rye or oat peptides.

Variations in antibody production or affinity among the same immunogen can be explained by biological variation between different hosts and initial immunogen processing by

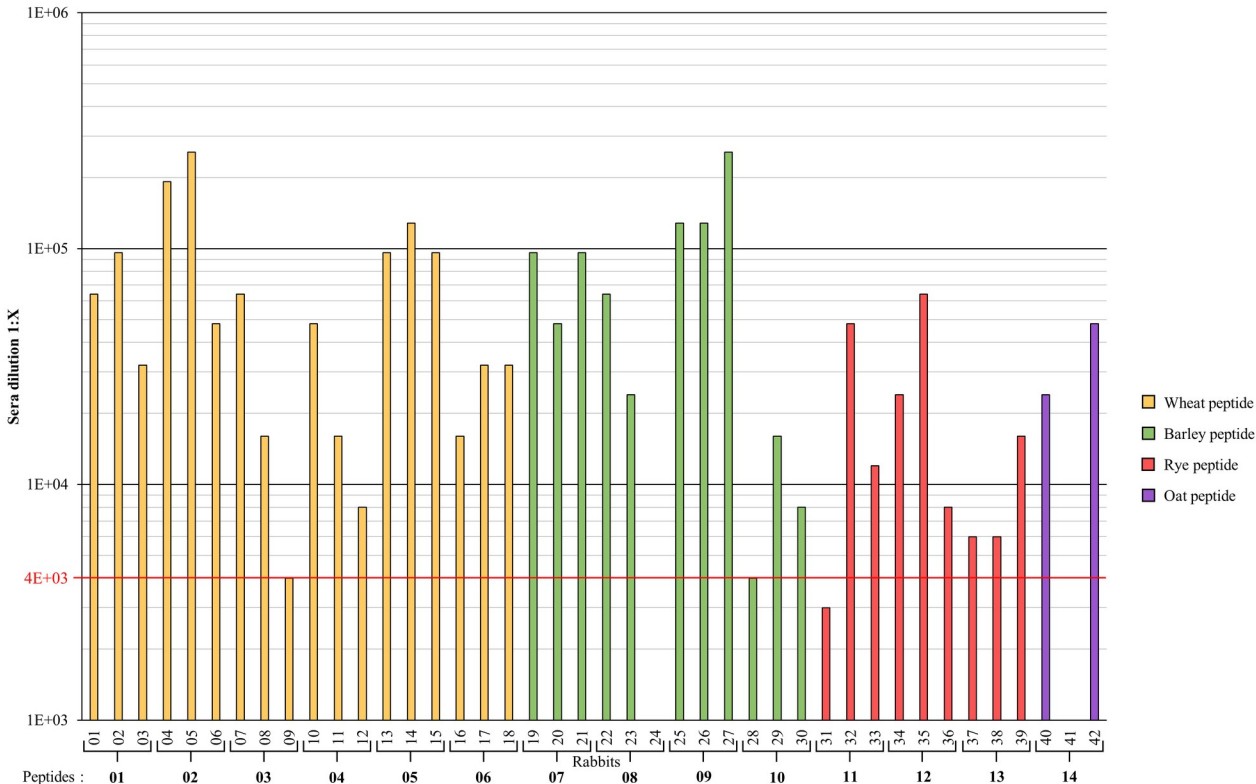

**Fig 2. Titer determination by indirect ELISA against 1µg/mL of BSA-(Mal-PEG4-NHS)-peptide (1–14).** Highest sera dilution triggering an O.D. value of ≈1.000 at 450 nm. Wheat peptide (01–06); Rabbits (01–18). Barley peptides (07–10); Rabbits (19–30). Rye peptides (11–13); Rabbits (31–39). Oat peptide (14); Rabbits (40–42). Rabbits 24 and 41 died before test bleed 1.

B-cells, which result in differences in raised pAbs [56, 57]. Titers below or equal to 1:4 000 were discarded since pAbs production was not considered strong enough for the purpose of this study: rabbit 9 (wheat), rabbit 28 (barley) and rabbit 31 (rye).

## Sensibility and specificity tests

Recognition of the GPT from which the synthetic peptides are sourced is not guaranteed as generation of antibodies is still not completely understood and is still based on empirical processes [46, 47, 53]. Screening results by indirect ELISA (Table 2) reveal low sensitivity for most of the pAbs against native and/or denatured GPT. These low recognition rates for certain prolamins contrast with the titers obtained against the BSA conjugates even if an approximately 10-fold molar excess was used for the prolamin coating over the peptides of the BSA conjugates, based on the MW of the different GPT.

For wheat, rabbits 1 to 6 and rabbit 18 present results above 1:8 000 against at least one of the two forms of GPT. On the other hand, sensitivity against native or denatured proteins collapsed for rabbit 7 to rabbit 17. For example, sera obtained with peptide 5 presented good results against the BSA bioconjugate, but negative or lower than 1:2 000 for the corresponding GPT. For barley, only the immunogen made with peptide 9 produced pAbs sensitive for the native form of the GPT. These results confirm the high titers obtained against the BSA bioconjugates (1:128 000 to 1:256 000). The sensitivities of all barley pAbs were not strong enough against the denatured form of the GPT with results equal to or lower than 1:2 000. For rye, the raised pAbs were not sensitive enough for either form of the GPT, which reflect the lower titer

**Table 2. Relative sensitivity tests by indirect ELISA.**

| Origin | Peptides | Rabbits | pAbs dilution [1:X] | | |
|---|---|---|---|---|---|
| | | | Synthetic peptides[a] | Native prolamins[b] | Denatured prolamins[c] |
| Wheat | 01 | 01 | 64000 | 20000 | 20000 |
| | | 02 | 96000 | 8000 | 8000 |
| | | 03 | 32000 | 20000 | 20000 |
| | 02 | 04 | 192000 | 4000 | 64000 |
| | | 05 | 256000 | 3000 | 16000 |
| | | 06 | 48000 | 3000 | 16000 |
| | 03 | 07 | 64000 | 0 | 0 |
| | | 08 | 16000 | 200 | 250 |
| | | 09 | 4000 | - | - |
| | 04 | 10 | 48000 | 0 | 0 |
| | | 11 | 16000 | 200 | 200 |
| | | 12 | 8000 | 250 | 200 |
| | 05 | 13 | 96000 | 0 | 0 |
| | | 14 | 128000 | 0 | 0 |
| | | 15 | 96000 | 1000 | 2000 |
| | 06 | 16 | 16000 | 750 | 500 |
| | | 17 | 32000 | 750 | 4000 |
| | | 18 | 32000 | 15000 | 40000 |
| Barley | 07 | 19 | 96000 | 500 | 500 |
| | | 20 | 48000 | 500 | 2000 |
| | | 21 | 96000 | 250 | 500 |
| | 08 | 22 | 64000 | 500 | 1000 |
| | | 23 | 24000 | 500 | 250 |
| | | 24 | 0 | - | - |
| | 09 | 25 | 128000 | 32000 | 1000 |
| | | 26 | 128000 | 2000 | 1000 |
| | | 27 | 256000 | 20000 | 500 |
| | 10 | 28 | 4000 | - | - |
| | | 29 | 16000 | 0 | 500 |
| | | 30 | 8000 | 250 | 250 |
| Rye | 11 | 31 | 3000 | - | - |
| | | 32 | 48000 | 1000 | 125 |
| | | 33 | 12000 | 0 | 0 |
| | 12 | 34 | 24000 | 250 | 125 |
| | | 35 | 64000 | 1000 | 100 |
| | | 36 | 8000 | 0 | 0 |
| | 13 | 37 | 6000 | 250 | 0 |
| | | 38 | 6000 | 0 | 0 |
| | | 39 | 16000 | 125 | 0 |
| Oat | 14 | 40 | 24000 | 12000 | 12000 |
| | | 41 | 0 | - | - |
| | | 42 | 48000 | 64000 | 64000 |

[a] Titers from [1μg/mL] BSA bioconjugate to synthetic peptides (1–14). O.D. ≈ 1.000 at 450 nm.

[b] Native prolamins [50μg/mL]. O.D. ≈ 1.000 at 450 nm.

[c] Denatured prolamins [50μg/mL]. O.D. ≈ 1.000 at 450 nm.

Shaded areas represent discarded sera that were discarded due to their low performances (dilutions ≤ 1:4000 to elicit a colorimetric response) for either type of prolamin (native or denatured).

**Table 3. Cross-reaction test by indirect ELISA.**

| Rabbits | N.[a] Wheat | D.[b] Wheat | N.[a] Barley | D.[b] Barley | N.[a] Rye | D.[b] Rye | N.[a] Oat | D.[b] Oat | Rice[a] | Split-pea[a] | Chickpea[a] | Millet[a] | Soy[a] |
|---------|------------|------------|--------------|--------------|-----------|-----------|-----------|-----------|---------|--------------|-------------|-----------|--------|
| 01 | +++ | +++ | + | +/- | - | - | + | +/- | - | - | - | - | - |
| 02 | ++ | ++ | + | + | +/- | - | + | - | n/a | n/a | n/a | n/a | n/a |
| 03 | +++ | +++ | + | + | ++ | ++ | +/- | + | n/a | n/a | n/a | n/a | n/a |
| 04 | ++ | +++ | - | - | - | - | - | - | - | - | - | - | - |
| 05 | + | ++ | - | - | - | - | - | - | - | - | - | - | - |
| 06 | + | ++ | +/- | - | - | - | +/- | - | - | - | + | - | - |
| 18 | +++ | +++ | +/- | - | +++ | ++ | +/- | +/- | - | - | - | - | - |
| 25 | +/- | - | +++ | - | - | - | + | + | - | - | - | - | - |
| 27 | - | - | ++ | - | - | - | +/- | - | - | - | - | - | - |
| 40 | - | - | - | - | - | - | +++ | +++ | +/- | +/- | +/- | - | - |
| 42 | - | - | - | - | - | - | +++ | +++ | - | - | - | - | - |

+++: 75–100% of the maximum O.D. signal by serum at 450nm.

++: 25–75% of the maximum O.D. signal by serum at 450nm.

+: 5–25% of the maximum O.D. signal by serum at 450nm.

+/-: $\leq$ 0.050 O.D. at 450nm. Value higher than blank cut-off value ($\alpha$ = 99.9%).

[a] N.: Native form of the prolamins. Coated at [50μg/mL]

[b] D.: Denatured form of the prolamins. Coated at [50μg/mL]

obtained with the related synthetic peptides. For oat, peptide 14 resulted in equally sensitive pAbs against either form of the GPT. Rabbit 42 gave pAbs even more reactive with the GPT (1:64 000) than the synthetic peptide (1:48 000).

These results were expected since the epitope occurrence is relatively higher in the BSA bio-conjugate, covered with synthetic peptides, than in the corresponding GPT. Several other factors can also explain these results. For native GPT, the targeted sequence is not necessarily present at the protein surface, limiting or preventing the fixation of the pAbs. For denatured GPT, even if the protein structure is disturbed, the targeted sequence could present a conformation that makes it unrecognizable by the pAbs [58]. It has been discussed earlier that hydrophilic peptides were selected since ideal antigenic haptens are hydrophilic and surface-oriented, making the epitopes exposed to the solvent [46–50]. However, hydropathy index of a peptide does not necessarily reflect the conformation of the corresponding stretch in the original protein. Indeed, secondary structure such as α-helices and β-sheets can hinder recognition by antibodies of the original protein, since linear peptides are able to assume a more random structure [58, 59]. These structures are inherently present in the native form of the prolamins, but also in the denatured state. It has been demonstrated that even in strong denaturant, some structures remain folded at some extent [60]. In milder conditions, such as in the buffer of the sensibility tests, the state of most denatured proteins would present considerable secondary structures [60]. Further analysis such as X-ray biocrystallography, NMR Spectroscopy or cryo-electron microscopy should be performed to better evaluate the repartition of these macromolecular structures in the native and denatured state of the GPT [61–63].

Only pAbs giving an O.D. of 1.000 at a dilution equal or higher than 1:4 000 for either native or denatured form of the prolamins were conserved. Then, only 11 sera were tested for potential cross-reactivity from the 37 tested (i.e. Rabbits: 01–06, 18, 25, 27, 40 and 42). Specificity tests results are shown in Table 3. Preliminary tests were conducted against wheat, barley, rye and oat prolamins. All the sera presented high specificity for their original prolamins, with the notable exception of rabbits 02 and 03, which were produced with a peptide originating from wheat but cross-reacted with all the prolamins independently of the cereal. For this

reason, these two sera were discarded from the second phase of the cross-reaction tests. Regarding the other sources of proteins, results of the cross-reaction testing show that some cross-reaction within multiple matrices occurred with rabbits 06 and 40. The pAbs from rabbit 06 cross-reacted with chickpea, while those from rabbit 40 did so with rice, split-pea and chickpea. Highly similar a.a. profiles between the different GPT and the ability of antibodies to recognize an extensive number of related epitopes, especially in pAbs, may explain these cross-reactions [64, 65]. Epitope mapping of the cross-reacting pAbs could point out the shared reactive sites [64, 66].

Several sera presented interesting results. One of these pAbs was specific to native and denatured wheat prolamins (rabbit 4: peptide 2 from α/β-gliadins). Another was specific to native barley prolamins (rabbit 25: peptide 9 from γ3-hordeins) with a residual reaction for oats, which could be discarded by immunoaffinity columns combined with epitope mapping [46, 66]. Finally, rabbits 40 and 42 (peptide 14 from avenins) presented specificity to both forms of oat prolamins. No specific pAbs were obtained against rye, but a pAbs with similar reactivity to native and denatured prolamins of wheat and rye was obtained (rabbit 18: peptide 6 from HMW-GS of wheat).

As no purification of the pAbs was done beforehand, rabbits serum proteins could have contributed to the residual signal, since the secondary antibody used in all the indirect ELISAs was a labelled goat anti-rabbit antibody. Moreover, the blanks used to determine the cut-off values were based on assay buffer, which does not contain rabbit serum to avoid nonspecific binding by the secondary antibody. Thus, this can slightly underestimate the threshold values when using Frey et al.'s (1998) statistical model, which might explain the presence of some negligible signals (+/-) [41]. In order to produce ELISA tests, a purification of the antibodies by protein A affinity would be necessary in order to overcome those signals [67].

## Conclusions

The immunization strategy based on synthetic peptides used in this study led to the identification of pAbs specific to native and denatured forms of wheat and oat prolamins, as well as pAbs specific to native barley prolamins. This strategy can therefore be regarded as a promising tool for the development of specific antibodies, thus facilitating compliance with Canadian regulations on the declaration of gluten sources in foods. In addition, this would directly benefit the wheat-allergic and coeliac populations by increasing their dietary options. Indeed, the R5, the method currently used for the detection of gluten and of traces of wheat in food, has a significant limitation, as it also measures barley and rye. This means that patients allergic to wheat have no option but to ban rye and barley from their diet. In addition, since R5 does not allow for detection of oats, the results of this study are also promising for European and Australian jurisdictions, which require declaration of oats on prepackaged food labels [68, 69]. Detection and quantification of the source of gluten can also complement current immunoassays by setting the issue of over and underestimation of gluten content. Further work should be conducted to raise an antibody specific to rye prolamins, since the one obtained in this study was equally specific to wheat and rye prolamins. Nonetheless, the latter could see applications in combination with the other specific pAbs to differentially determine the source of gluten [70]. Further developments for the creation of monoclonal antibodies based on the immunization strategy presented here are planned, as well as further experimentations against processed food products. This ongoing work is expected to overcome the residual cross-reactivity between the various GPT, since pAbs are characterized by a lower specificity, as compared to monoclonal antibodies (i.e. epitope-specific) [46] and could pave the way for the production of new ELISA assays.

## Supporting information

**S1 Table. Consensus sequences obtained from multiple alignments sequences (MSA) and compilation.**
(DOCX)

## Acknowledgments

Many thanks to Virginie Barrere who helped with the PCR protocols, Anne-Catherine Huet who provided technical assistance for the immunoassays, Sylvia Dominguez who helped with the language correction and Semican International Inc. and Avena Foods who generously supplied the grains used in this study.

## Author Contributions

**Conceptualization:** David Poirier, Jérémie Théolier, Samuel Benrejeb Godefroy.

**Data curation:** David Poirier.

**Formal analysis:** David Poirier, Riccardo Marega.

**Funding acquisition:** Jérémie Théolier, Nathalie Gillard, Samuel Benrejeb Godefroy.

**Methodology:** David Poirier, Jérémie Théolier, Riccardo Marega, Samuel Benrejeb Godefroy.

**Project administration:** Jérémie Théolier, Samuel Benrejeb Godefroy.

**Resources:** Jérémie Théolier, Samuel Benrejeb Godefroy.

**Software:** David Poirier.

**Supervision:** Jérémie Théolier, Samuel Benrejeb Godefroy.

**Validation:** Samuel Benrejeb Godefroy.

**Visualization:** David Poirier, Samuel Benrejeb Godefroy.

**Writing – original draft:** David Poirier.

**Writing – review & editing:** David Poirier, Jérémie Théolier, Riccardo Marega, Philippe Delahaut, Nathalie Gillard, Samuel Benrejeb Godefroy.

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
