## [Decision Letter · Decision Letter 0]

22 Jun 2021

PONE-D-21-12474

Evaluation of the discriminatory potential of antibodies created from synthetic peptides derived from wheat, barley, rye and oat gluten

PLOS ONE

Dear Dr. Poirier

Thank you for submitting your manuscript to PLOS ONE. After careful consideration, we feel that it has merit but does not fully meet PLOS ONE’s publication criteria as it currently stands. Therefore, we invite you to submit a revised version of the manuscript that addresses the points raised during the review process.

We look forward to receiving your revised manuscript.

Kind regards,

Patrizia Restani, Ph.D.

Academic Editor

PLOS ONE

When submitting your revision, we need you to address these additional requirement.

Additional Editor Comments (if provided):

The paper must be improved according to the reviewers' comments.

Reviewers' comments:

Reviewer's Responses to Questions

**Comments to the Author**

1. Is the manuscript technically sound, and do the data support the conclusions?

Reviewer #1: Yes

Reviewer #2: Yes

2. Has the statistical analysis been performed appropriately and rigorously? 

Reviewer #1: Yes

Reviewer #2: N/A

3. Have the authors made all data underlying the findings in their manuscript fully available?

Reviewer #1: Yes

Reviewer #2: Yes

4. Is the manuscript presented in an intelligible fashion and written in standard English?

Reviewer #1: Yes

Reviewer #2: Yes

5. Review Comments to the Author

Reviewer #1: The paper by David Poirier et al. presents interesting results about the development of new antibodies targeting discrimination of gluten sources using synthetic peptides as immunization strategy. The aim is to provide possible analytical solutions to solve the issue of discriminating gluten sources, as requested by the Canadian regulations on this matter. Moreover, a possible contribution to improve the actual immunoenzymatic diagnostic tools for gluten quantification in foods is foreseen.

The approach based on synthetic peptides bioconjugated to a protein carrier as immunization strategy, to elicit antibodies in rabbit, is up to date and very interesting and the procedure and results are well presented and discussed.

As a matter of fact, although very common, easy to use and low cost, currently commercially available ELISA tests show very often inconsistency among different producers and underestimation or overestimation problems about gluten content, especially in processed and fermented foods.

This is frequently linked to recovery problems, limited ability to detect hydrolyzed gluten (with underestimation of its content), interferences such as high salt content that could lead to gliadin precipitation, difficulties in optimizing extraction from highly processed food such as pasta or biscuits.

For these reasons, several alternative approaches are currently being studied such as the use of competitive or multiplex assay or alternative analytical methodologies such as LC-MS/MS.

Thus, in order to better define the impact of the results obtained in this research work, I would suggest comments or discussion on the following points which have to be better clarified:

1) Apart from cereal flour, is the approach potentially applicable also to processed and final food products (heat-treated food, fermented food such as beer) ? This is an important point in order to evaluate the possible applicability of the approach for monitoring of “gluten-free” food products monitoring as far as gluten content quantification.

2) It is not clear from the discussion how this type of discrimination could also complement current immunoassays by settling the issue of over and underestimation of gluten content, thus improving the safety of food intended to CD and wheat-allergic patients, because it is not clear if this approach could target also gliadins in processed foods. Is it the test on denatured proteins sufficient to foreseen such a possible application?

3) LC-MS/MS have been already used to differentiate gluten sources. The possibility to set up an efficient complementary ELISA methodology is very interesting, but it is not clear from the presented results if this approach could be efficacious also on processed food obtained from flour or cereal ingredients.

Reviewer #2: In this paper, the development of new antibodies targeting discrimination of gluten sources was conducted using synthetic peptides as immunization strategy.

The resulting polyclonal antibodies (pAbs) successfully discriminated wheat, barley and oat prolamins during indirect ELISA assessments.

Some minor observations to be corrected before publication

ml  mL

poor quality of Fig 1 - page 33

poor quality of Fig 2 - page 35 - recommendation is to use colors and legend for a better understanding

6. PLOS authors have the option to publish the peer review history of their article (what does this mean?). If published, this will include your full peer review and any attached files.

Reviewer #1: **Yes: **Gianni Galaverna

Reviewer #2: No

---

## [Author Response · Author response to Decision Letter 0]

16 Jul 2021

Dear Professor Restani,

We are pleased to submit our revised manuscript “Evaluation of the discriminatory potential of antibodies created from synthetic peptides derived from wheat, barley, rye and oat gluten” for consideration for publication in PLOS one. please acknowledge the following justifications for the raised points by the reviewers.

Reviewer #1:

1) Apart from cereal flour, is the approach potentially applicable also to processed and final food products (heat-treated food, fermented food such as beer) ? This is an important point in order to evaluate the possible applicability of the approach for monitoring of “gluten-free” food products monitoring as far as gluten content quantification.

It is expected that this approach will work for heated products. For fermented products, we cannot give a firm answer. In our opinion, it is too premature to talk about applicability for processed products as our work was not designed for this purpose. The object of the present manuscript was to validate a protocol for raising antibodies based on synthetic peptides in order to obtain specific antibodies.

As mentioned at lines 348 to 351 of the manuscript, further experimentations are planned with monoclonal antibodies. These experimentations will include the production of ELISA tests and a validation of the developed ELISA tests against multiple food matrices with incurred gluten. Clarifications have been made to line 349-350 of the manuscript.

2) It is not clear from the discussion how this type of discrimination could also complement current immunoassays by settling the issue of over and underestimation of gluten content, thus improving the safety of food intended to CD and wheat-allergic patients, because it is not clear if this approach could target also gliadins in processed foods. Is it the test on denatured proteins sufficient to foreseen such a possible application?

Over and underestimation of gluten in food products is the result of the way gluten is quantified (it is a side effect of the R5 test). The main factor for overestimation and underestimation is the origin of gluten. (Lines 38-40) The production of specific antibodies would resolve this issue because it will be possible to know the origin of gluten. Clarifications have been added in the text line 47-48. 

As mentioned at lines 42 to 47 of the manuscript: “The use of a reference material for each grain, namely wheat, barley, rye and oats, as a calibration standard makes it possible to correct or at least reduce this variability”. However, “calibration is often impossible in food analysis, since the source of gluten, if present due to cross-contamination, is usually unknown”. Therefore, being able to identify the source of gluten in a food sample would help in selecting the appropriate calibration standards. 

3) LC-MS/MS have been already used to differentiate gluten sources. The possibility to set up an efficient complementary ELISA methodology is very interesting, but it is not clear from the presented results if this approach could be efficacious also on processed food obtained from flour or cereal ingredients.

We agreed that LC-MS/MS is a way better tool than ELISA to differentiate gluten sources. However, LC-MS/MS are still marginal in food industries especially for the detection of gluten because the reference method is ELISA. The ultimate goal of this project is to develop tools that can be used by food industries, and considering their limited financial resources, the development of a new ELISA test seemed to be a good option. However, as mentioned at lines 53-54: “The aim of this study was to create new antibodies capable of distinguishing between different sources of gluten.”

ELISA tests need to be validated against several matrices before commercialization. We are currently in the process to produce these kits so talking about processed products is too early (see response to question 1).

Reviewer #2:

All “ml” unit notations have been changed to “mL”.

Fig 2 quality has been improved and colors have been added to the figure and legend for a better understanding.

Warm regards,

David Poirier 

Université Laval

---

## [Decision Letter · Decision Letter 1]

2 Sep 2021

Evaluation of the discriminatory potential of antibodies created from synthetic peptides derived from wheat, barley, rye and oat gluten

PONE-D-21-12474R1

Dear Dr. David Poirier

We’re pleased to inform you that your manuscript has been judged scientifically suitable for publication and will be formally accepted for publication once it meets all outstanding technical requirements.

Kind regards,

Patrizia Restani, Ph.D.

Academic Editor

PLOS ONE

Additional Editor Comments (optional):

The revision met the reviewer's requests.

Reviewers' comments:

Reviewer's Responses to Questions

**Comments to the Author**

1. If the authors have adequately addressed your comments raised in a previous round of review and you feel that this manuscript is now acceptable for publication, you may indicate that here to bypass the “Comments to the Author” section, enter your conflict of interest statement in the “Confidential to Editor” section, and submit your "Accept" recommendation.

Reviewer #1: All comments have been addressed

2. Is the manuscript technically sound, and do the data support the conclusions?

Reviewer #1: Yes

3. Has the statistical analysis been performed appropriately and rigorously? 

Reviewer #1: Yes

4. Have the authors made all data underlying the findings in their manuscript fully available?

Reviewer #1: Yes

5. Is the manuscript presented in an intelligible fashion and written in standard English?

Reviewer #1: Yes

6. Review Comments to the Author

Reviewer #1: Dear author,

I'm satisfied by your explanation and integration of the manuscript. I think the paper is now suitable to be published in PLOS ONE.

7. PLOS authors have the option to publish the peer review history of their article (what does this mean?). If published, this will include your full peer review and any attached files.

Reviewer #1: **Yes: **Gianni Galaverna

---

## [Editor Report · Acceptance letter]

13 Sep 2021

PONE-D-21-12474R1 

Evaluation of the discriminatory potential of antibodies created from synthetic peptides derived from wheat, barley, rye and oat gluten 

Dear Dr. Poirier:

I'm pleased to inform you that your manuscript has been deemed suitable for publication in PLOS ONE. Congratulations! Your manuscript is now with our production department. 

Kind regards, 

on behalf of

Professor Patrizia Restani 

Academic Editor

PLOS ONE